# When women eat last: Discrimination at home and women's mental health

**Payal Hathi** [1,2]* *, **Diane Coffey**[2,3,4]*, **Amit Thorat**[2,5]*, **Nazar Khalid**[2,6]*

**1** Departments of Sociology & Demography, University of California, Berkeley, Berkeley, California, United States of America, **2** r.i.c.e., a Research Institute for Compassionate Economics, India, **3** Population Research Center, University of Texas at Austin, Austin, Texas, United States of America, **4** Indian Statistical Institute, Delhi Centre, Delhi, India, **5** Centre for the Study of Regional Development, Jawaharlal Nehru University, Delhi, India, **6** Department of Demography, University of Pennsylvania, Philadelphia, Pennsylvania, United States of America

☯ These authors contributed equally to this work.
* phathi@berkeley.edu

## Abstract

The 2011 India Human Development Survey found that in about a quarter of Indian households, women are expected to have their meals after men have finished eating. This study investigates whether this form of gender discrimination is associated with worse mental health outcomes for women. Our primary data source is a new, state-representative mobile phone survey of women ages 18–65 in Bihar, Jharkhand, and Maharashtra in 2018. We measure mental health using questions from the World Health Organization's Self-Reporting Questionnaire. We find that, for women in these states, eating last is correlated with worse mental health, even after accounting for differences in socioeconomic status. We discuss two possible mechanisms for this relationship: eating last may be associated with worse mental health because it is associated with worse physical health, or eating last may be associated with poor mental health because it is associated with less autonomy, or both.

**Data Availability Statement:** The data underlying the results presented in the study are available from: https://riceinstitute.org/data/social-attitudes-research-india-sari-data/. Documentation of data are available from: http://riceinstitute.org/data/sari-

## 1. Introduction

Women and girls in India face many forms of discrimination throughout the life course. Specific forms of discrimination include sex selection, not being able to go to school, disapproval of working outside the home, having limited decision-making power, and denial of property ownership and control over money. Gender inequality has important consequences for women themselves, and also for their families and communities. For example, women's lower social status is associated with poorer child health outcomes, both within India [1], and in cross-national studies across the developing world [2]. While recent studies have explored gender inequality in health in India along dimensions such as childhood immunization [3] and health expenditure [4], other aspects of gender discrimination related to health, particularly mental health, have received much less attention.

An important form of gender discrimination in India occurs in the ways in which food is distributed within households. For example, girl babies are breastfed for shorter periods than boy babies [5], girl children are given less and worse food than boy children [6, 7], and

dataset-documentation/. The authors collected and cleaned the data themselves. The version of the data used in this analysis is the same data that is publicly available, and required no special access privileges to the data.

**Funding:** Data collection was supported by a grant from the Bill & Melinda Gates Foundation (# OPP1125318) to r.i.c.e., and analysis was supported by a grant from the International Growth Centre (# 1-VCH-VINC-VXXXX-35114). Research was supported by a National Institute of Child Health and Human Development Training grant at the University of California, Berkeley [T32HD007275] (PH), and by a grant awarded to the Population Research Center at The University of Texas at Austin by the Eunice Kennedy Shriver National Institute of Child Health and Human Development [P2CHD042849] (DC). The content is solely the responsibility of the authors and does not necessarily represent the official views of the National Institutes of Health. The funders had no role in study design, data collection and analysis, decision to publish, or preparation of the manuscript.

**Competing interests:** The authors have declared that no competing interests exist.

women, despite doing almost all of the cooking, are often expected to eat last [8]. When women eat their meals after men, they often eat leftover food that is of lower quality than what they would consume if men and women ate together [9]. The first nationally representation quantification of this discriminatory practice comes from the 2005 India Human Development Survey (IHDS). In 2005, 36% of women said that, in their households, women ate after men [10]. The IHDS is a panel study that revisited the same households in 2011. There was some change between the two survey rounds; in 2011, about one-quarter of women in India reported that women eat after men in their households [11].

Although some families might explain the practice of women eating last as an appropriate way for them to show respect for their husbands and in-laws, or as a practical way to ensure that everyone other than the cook can eat hot *roti* (flat breads), this practice has observable negative consequences for women's nutrition. Coffey & Hathi [12] document that women are more likely to be underweight than men, at all adult ages. Using IHDS data, Coffey et. al. [8] show that, at all levels of per capita household consumption, women who live in households in which women eat last are more likely to be underweight than women who live in households in which they do not eat last. National Family Health Survey (NFHS) 2015–16 data show that undernutrition is widespread nationally: close to one-quarter of women in India are underweight, with a body mass index score of less than $18.5 \, \mathrm{kg/m^2}$. Undernutrition has negative consequences for the women who experience it, as it leads to lower levels of energy [13], and more frequent sickness due to compromised immunity [14]. Undernutrition also has important intergenerational health consequences: women who are underweight before pregnancy are more likely to have babies who are small, and more likely to have their newborns die within the first month of life [15].

Does the practice of women eating last also affect women's mental health? Although the medical literature shows that being underweight can impact concentration, decision-making, and mood, it has not previously been possible to explore the relationship between women eating last and their mental health because no dataset combines a question on this discriminatory practice with measures of mental health. For example, the IHDS includes detailed data on discrimination against women, including the question of whether women eat last, but does not include mental health, and the WHO Study on Global Ageing and Adult Health (SAGE) asks about mental health, but not gender discrimination. To our knowledge, no prior dataset asks about both.

This paper draws on data from Social Attitudes Research, India (SARI), a representative mobile phone survey, to investigate how poor mental health correlates with discrimination against women. We first show that there is a large gap in mental health between women who do and do not eat last. We note that eating last is highly correlated with both a woman's education and her household's economic status, both factors which are known to independently predict mental health. Education may be related to mental health for several reasons: low educational attainment could be an indication of childhood adversity, or a proxy for social position and or lack of opportunity more broadly [16]. Economic status may also be associated with poor mental health: while levels of poverty may not clearly predict worse mental health, the poor are more likely to experience adverse events in their lives that lead to greater insecurity and hopelessness [17, 18].

We use two empirical strategies to show that, although correlations between eating last and education, and eating last and economic status can explain some of the gap, the difference in mental health between women who do and do not eat last cannot be entirely explained by these differences. As a contrast to the literature that shows that poor mental health may be caused by poor physical health [18–20], we explore women's autonomy as one possible pathway through which eating last may be associated with poor mental health, even net of

socioeconomic controls. We analyze two different measures of women's autonomy, which has been shown in other contexts to cause poor mental health because of the stresses associated with a lack of control over the circumstances of one's life [21]. It is possible that both physical health and women's autonomy influence women's mental health at the same time. We present suggestive evidence from a collage of data sources that both of these channels are likely important in explaining why eating last is associated with poor mental health.

The rest of the paper proceeds as follows. The Background section provides background on gender and mental health in India. The Methods section describes the datasets and variables used in the analysis, as well as our empirical strategy. The Results section presents our analyses: first, we present two empirical strategies to understand the extent to which socioeconomic and demographic characteristics can explain the gap in mental health between women who do and do not experience discrimination in the form of eating last. Next, we analyze data from nationally representative data on women's autonomy to explore whether women's autonomy mediates the relationship between gender discrimination and mental health. We then discuss our findings and conclude.

## 2. Background: Gender and mental health in India

Most studies explore gendered outcomes in mental health in the context of high-income countries because many surveys that include questions on mental health have been developed and used only in these contexts [22–24]. This has left mental health among populations in low-income countries less well-understood. Studies of women's mental health that are conducted in poor countries often focus on reproductive health, emphasizing the experience of motherhood, or are conducted for the purpose of assessing community-level prevalence of mental health disorders [25, 26]. Instead, this study considers women's mental health more broadly, at the population level in three states in India, where strong patriarchal norms mean that women face discrimination throughout the life course.

Although no prior research has explored the relationship between eating last and women's mental health, prior research has begun to explore the role of patriarchy in putting women at greater risk than men for mental health disorders in India. In a study assessing various risk factors for common mental disorders (CMDs), which include anxiety and depressive disorders, Patel et al. [27] find that gender disadvantage, including low autonomy in decision-making, lack of social integration, and physical and sexual violence within marriage, is strongly associated with the prevalence of CMDs. Similarly, Das et al. [25] find that some of the difference in levels of poor mental health between men and women can be explained by women's greater sensitivity to adverse reproductive outcomes, including the death of a child, because this may represent a loss of a woman's expected role as a mother in society.

However, gendered patterns of mental health are not always straight forward. For example, Coffey & Gupta [28] find that women in India report greater happiness than men, despite severe gender discrimination. They find that this gap exists among young people, but not older people. They posit that the gap between young men's and women's happiness could arise if younger women perform happiness as a part of their expected gender roles. Similarly, Yim & Mahalingam [29] find that in Indian states with high male to female sex ratios, which indicate clear and extreme discrimination against women, women who endorsed culturally idealized gender roles experienced less anxiety, despite the fact that such roles constrain women's own autonomy. Vindhya [30] discusses the meaning of autonomy in a society like India's in which women's self-esteem is often tied to the fulfillment of their idealized roles in relation to their spouses, children, or other family members, rather than being about their individual needs. Finally, Rawat [31] explains that if women remain confined by patriarchal social norms,

increasing levels of education and workplace participation alone may not be sufficient to lead to greater wellbeing for women. Given the subtleties in these findings, it is not clear in advance whether eating last will clearly predict mental health or not.

## 3. Methods

### 3.1. Data

This study analyzes two datasets. The first is the Social Attitudes Research, India (SARI) survey, a mobile phone survey that collects data from adults ages 18 to 65 in Indian states. It measures attitudes towards marginalized groups and public opinion on current policies. This survey is unique because it contains questions both about mental health and about women's status, whereas prior surveys collected data on only one or the other. The second dataset we analyze, the India Human Development Survey (IHDS) 2005 and 2012 is a nationally representative panel survey that collected extensive information on women's status and autonomy, but did not ask about mental health.

For both datasets, we restrict our analyses to ever-married women, aged 25 and above. Our primary reason for doing so is that women's social status in India often changes dramatically after marriage, with much stricter enforcement of patriarchal gender roles for married women. By age 25, close to 95% of women are married.

**3.1.1. Social Attitudes Research, India (SARI).** SARI has collected data in Delhi, Uttar Pradesh, Rajasthan, Mumbai, Bihar, Jharkhand, and Maharashtra. However, this study focuses on data collected in 2018 in Bihar, Jharkhand, and Maharashtra because mental health questions were asked only in these three states. Sample sizes and response rates to the survey and mental health questions are given in Table 1 below.

SARI uses random digit dialing in order to recruit representative samples of respondents in each state. India's Department of Telecommunications issues 5-digit, location-specific "series" to phone companies to be used as the first 5 digits of the phone numbers that they sell to consumers. In proportion to the number of subscribers that belong to each phone company, SARI then combines these 5-digit "series" with 5 randomly generated digits to form 10-digit phone numbers. Interviewers call these phone numbers in a random order.

Interviewers speak to respondents of the same sex in order to reduce social desirability bias and maximize respondent comfort. Once a respondent of the correct sex agrees to participate, they are asked to list all adults of their sex in the household. Survey respondents are selected randomly from the household listing by Qualtrics software to ensure (1) that even individuals who do not own their own mobile-phones are eligible to be interviewed, and (2) that even the

**Table 1. Survey and SRQ sample sizes and response rates.**

| | Overall survey | | SRQ Questionnaire | |
|---|---|---|---|---|
| | Sample size | Response rate | Sample size | Response rate |
| Bihar (2018) | 3438 | 19% | 1619 | 93% |
| Jharkhand (2018) | 1009 | | 500 | 89% |
| Maharashtra (2018) | 1666 | 25% | 784 | 95% |

Note: Survey response rates are calculated the number surveys in which a respondent answered at least a third of the questions, divided by the number of mobile numbers that were valid (as opposed to nonexistent, switched off, or not available) when they were first called. Response Rates for Bihar and Jharkhand cannot be calculated separately because Bihar and Jharkhand mobile numbers are pooled into the same mobile circle by the Telecom Regulatory Authority of India. State of residence is only known for individuals who began the survey, but not for every valid phone number called. Response rates for the SRQ questionnaire include individuals who answered all six questions.

least educated adults, who may be less likely to participate in a phone survey, are represented in our sample.

Since individuals from some demographic groups are more likely to respond to the survey than others, we follow common survey practice and construct weights using data from the 2011 India Census. Weights account for the intersection of sex, place (i.e. urban/rural), education, and age, and allow us to construct representative samples of adults in each state population. More details about data collection, survey design, and analysis can be found in Coffey et al. [8], Hathi et al. [32], and online at http://riceinstitute.org/data/sari-dataset-documentation/.

**3.1.2. India Human Development Survey (IHDS).** The India Human Development Survey (IHDS) is a nationally representative panel survey of over 41,000 rural and urban households, conducted using face-to-face surveying in 2005 and 2012 [10, 11]. In addition to questions about health and education, marriage, household assets and poverty, the IHDS also included several questions on women's social status and autonomy. In this article, we focus on IHDS questions from 2012 about decision-making power in the household, asked to ever-married women.

**3.1.3. SARI and IHDS questions.** SARI randomly assigned respondents to be asked one of two sets of commonly used mental health questions, an adapted Kessler-6 Questionnaire and an adapted Self-Reporting Questionnaire (SRQ). The adaptations to these questionnaires facilitated their use in a mobile survey as opposed to an in-person survey. For this study, we omit results for the Kessler-6 questions because the proportion of respondents who answered all Kessler-6 questions was more than 10 percentage points lower than for the SRQ questions, across all states. We thus analyze only results from women who were assigned to answer the SRQ, which had a higher response rate.

The SRQ was developed by the WHO in order to be implemented in the developing country context by primary health workers with limited training in screening for and identifying psychiatric symptoms [33]. It includes 20 questions that are easy to understand, and answerable with simple "yes" or "no" responses. SRQ questions have been shown to be able to detect common mental disorders with reasonable accuracy, and several studies have validated adaptations of the SRQ across cultural settings [34–38]. We chose to ask six questions rather than 20 for three reasons. First, the format of a mobile phone survey did not allow for a lengthy 20 question screening. Second, respondents in the pilot surveys reported difficulty differentiating among some of the SRQ questions: they thought we were repeating the same question more than once. This led us to choose questions that were different from one another. We chose six questions to match the number of questions in the Kessler questionnaire. We selected questions that focused on physical symptoms to contrast the focus on emotional questions in the Kessler questionnaire. The adapted SRQ was validated during pilot testing to ensure that respondents understood what was being asked. Coffey et al. (2020) [28] report on the design of the mental health portion of the SARI survey in greater detail [39]. The adapted SRQ that was used in SARI is given in Table 2 below. S1 Table lists the full set of 20 SRQ questions as they appear in the WHO's User's Guide to the Self Reporting Questionnaire [33].

As recommended by the SRQ User Guide, interviewers introduced the SRQ questions with the following text: "In the next few questions, I will ask you about the sadness or problems you may have faced in the last 30 days. If something like this happened in the last 30 days, say yes. If this did not happen in the last 30 days, say no. Now I will ask you questions one-by-one."

Some of SARI's questions were modeled off of IHDS questions to be able to confirm data quality in SARI [8]. In particular, the question about eating order analyzed in this study, was asked using the same wording in both SARI and the IHDS.

**Table 2. Mental health and autonomy questions analyzed (SARI & IHDS).**

| SARI | IHDS |
|---|---|
| **Mental health:** | |
| SRQ: | |
| 1. Is your appetite poor? | |
| 2. Do you have trouble sleeping? | |
| 3. Do you have trouble thinking clearly? | |
| 4. Do you find it difficult to make decisions? | |
| 5. Has the thought of ending your life been on your mind? | |
| 6. Do you feel tired all the time? | |
| **Autonomy:** | |
| 1. When your family eats lunch or dinner, do the women usually eat with the men? Or do the women usually eat first? Or do the men usually eat first? | 1. When your family takes the main meal do women usually eat with the men? Do women eat first by themselves? Or do men eat first? |
| 2. When you want to go outside alone somewhere near your home, such as to visit a neighbor, do you need to ask your husband or family, or do you just tell them and go? | 2. Please tell me who in your family has the most say in the decision: |
| | a. What to cook on a daily basis? |
| | b. Whether to buy an expensive item such as a TV or fridge? |
| | c. How many children you have? |
| | d. What to do if you fall sick? |
| | e. Whether to buy land or property? |
| | f. How much money to spend on a social function such as marriage? |
| | g. (if respondent has children) What to do if a child falls sick? |
| | h. To whom your children should marry? |

In order to understand the mechanisms for how and why gender discrimination may influence mental health, we use IHDS questions on women's decision-making power in the household to explore the correlations between the eating last variable and other indicators of women's status. The questions used from the IHDS are also shown in Table 2.

## 3.2. Empirical strategy

People who experience social discrimination may also experience economic and educational disadvantage. Therefore, if we see correlations between discrimination and mental health in our analyses, it may be attributable to these factors, which are also associated with poor mental health. Therefore, our empirical strategy is designed to measure the association between eating last and mental health net of differences in household economic status and women's education that exist between households in which women eat last and those in which they do not. We note, however, that even if differences in women's education can explain part of the association between women eating last and mental health does not mean that gender discrimination is not an ultimate cause of poor mental health. Indeed, households with more gender discrimination will likely both discourage women from pursuing education and enforce the practice of women eating last.

**3.2.1. Descriptive statistics.** We first show descriptive statistics of the differences in socio-economic characteristics and mental health between women who live in households in which women eat last, and those households in which they do not eat last. We report the fraction of women who say that they experienced a particular symptom of poor mental health in the 30

days prior to the survey. Additionally, we describe the population in terms of age, asset ownership, years of education, rural residence, and state of residence. We also include caste group and whether respondents are Muslim. Caste and religion are two important dimensions of social disadvantage in India.

Several prior community studies ask whether women whose households lack a toilet or latrine, and who therefore defecate in the open, experience stress that contributes to poor mental health [40, 41]. Sahoo et al. [42] find that environmental factors (i.e. long distances or unclean facilities), social factors (i.e. insufficient privacy or conflicts over scarce sanitation infrastructure), and fears of sexual violence increased sanitation-related psychosocial stress among women in Orissa. They also find that women have little ability to modify these difficult circumstances. Because lack of a toilet or latrine may lead to mental distress for women in the states we study as well, we include latrine ownership as an important predictor variable. Analyses that control for latrine ownership include only data from rural women: SARI only asked rural residents about latrine ownership because India's open defecation is concentrated in rural areas. Another reason that we did not ask urban residents about latrine ownership and use was that, in piloting, they sometimes became upset at the suggestion that they might defecate in the open.

**3.2.2. Non-parametric reweighting.** Following non-parametric reweighting standardization techniques used by DiNardo et al. [43], Geruso [44], and Coffey [45], we estimate counterfactual total SRQ scores for women who eat last, reweighting their cumulative distribution function by forcing the distributions of their socioeconomic characteristics to match those of women who do not eat last. This tells us what the total SRQ score of women who eat last would look like if they had the same distribution of asset wealth and educational attainment as women who do not eat last.

In order to produce this counterfactual distribution, we estimate the following reweighting function:

$$\psi(x) = \frac{f(x|\text{women who do not eat last})}{f(x|\text{women who eat last})} \qquad \text{(Eq 1)}$$

where $x$ is a vector of indicators for the intersections of the four educational attainment categories and six asset ownership categories described in Table 3, and $f$ is the probability density function. By reweighting over 24 education by asset ownership bins, this function changes the distribution of the observed education and asset characteristics of women who eat last so that it matches the distribution of women who do not eat last. To calculate the reweighting function, for each asset by education bin, we divide the fraction of women who do not eat last in that bin out of the total sample, by the fraction of women who do eat last in that bin out of the total sample. Each individual is multiplied by her corresponding reweighting function, such that a counterfactual distribution is computed for a counterfactual population of women who eat last, but whose education and asset ownership match that of women who do not eat last.

The counterfactual reweighted distribution of total SRQ score $m$ is

$$m = \frac{\sum_i \psi(x_i) w_i m_i}{\sum_i \psi(x_i) w_i} \qquad \text{(Eq 2)}$$

where $m_i$ is the total SRQ of person $i$; $x_i$ is the education by asset bin of person $i$, and $w_i$ is the survey sampling weight of person $i$. Intuitively, this means that if women who eat last own fewer assets, on average, than women who do not eat last, the reweighting function will up-weight wealthier women who eat last, and down-weight poorer women who eat last. And if

**Table 3. Summary statistics.**

| | Total | | women who eat last | | women who do not eat last | |
|---|---|---|---|---|---|---|
| | Proportion/mean | 95% CI | mean | 95% CI | mean | 95% CI |
| **Self reported mental health symptoms in past 30 days:** | | | | | | |
| Felt lack of appetite | 0.42 | [0.37, 0.47] | 0.50 | [0.44, 0.57] | 0.36 | [0.30, 0.43] |
| Had trouble sleeping | 0.43 | [0.38, 0.48] | 0.59 | [0.52, 0.65] | 0.33 | [0.28, 0.39] |
| Had trouble thinking clearly | 0.40 | [0.35, 0.45] | 0.54 | [0.47, 0.60] | 0.31 | [0.26, 0.37] |
| Felt tired all the time | 0.19 | [0.16, 0.23] | 0.24 | [0.19, 0.30] | 0.16 | [0.12, 0.21] |
| Had trouble making decisions | 0.65 | [0.60, 0.70] | 0.75 | [0.68, 0.80] | 0.59 | [0.51, 0.66] |
| Contemplated suicide | 0.44 | [0.40, 0.50] | 0.53 | [0.46, 0.60] | 0.39 | [0.33, 0.46] |
| **Predictor variables** | | | | | | |
| Mean age | 39.8 | [38.60, 41.05] | 39.0 | [37.41, 40.68] | 40.3 | [38.62, 42.05] |
| Mean # assets (out of 5) | 2.8 | [2.6, 2.9] | 2.3 | [2.1, 2.5] | 3.1 | [2.9, 3.3] |
| Asset categories | | | | | | |
| 0 | 0.13 | | 0.17 | | 0.10 | |
| 1 | 0.15 | | 0.20 | | 0.12 | |
| 2 | 0.16 | | 0.19 | | 0.15 | |
| 3 | 0.15 | | 0.17 | | 0.15 | |
| 4 | 0.20 | | 0.17 | | 0.22 | |
| 5 | 0.20 | | 0.10 | | 0.27 | |
| Mean years of education | 3.9 | [3.5, 4.3] | 2.6 | [2.2, 3.0] | 4.8 | [4.2, 5.4] |
| Years of education categories | | | | | | |
| No education | 0.50 | | 0.61 | | 0.43 | |
| 1 to 8 | 0.30 | | 0.28 | | 0.30 | |
| 9 to 12 | 0.14 | | 0.08 | | 0.17 | |
| More than 12 | 0.06 | | 0.03 | | 0.09 | |
| Caste group | | | | | | |
| Brahmin | 0.04 | | 0.04 | | 0.04 | |
| General | 0.33 | | 0.24 | | 0.39 | |
| Other Backward Caste | 0.36 | | 0.45 | | 0.31 | |
| Dalit | 0.20 | | 0.21 | | 0.20 | |
| Adivasi | 0.05 | | 0.04 | | 0.05 | |
| Other | 0.01 | | 0.01 | | 0.01 | |
| Rural resident | 0.72 | [0.67, 0.77] | 0.82 | [0.75, 0.87] | 0.67 | [0.60, 0.73] |
| Latrine ownership (rural only) | 0.57 | [0.52, 0.63] | 0.49 | [0.42, 0.56] | 0.64 | [0.55, 0.71] |
| Muslim | 0.11 | [0.09, 0.14] | 0.11 | [0.08, 0.15] | 0.11 | [0.08, 0.14] |
| State | | | | | | |
| Bihar | 0.40 | | 0.63 | | 0.25 | |
| Jharkhand | 0.05 | | 0.05 | | 0.05 | |
| Maharashtra | 0.55 | | 0.33 | | 0.70 | |
| n | 1218 | | 519 | | 697 | |

Note: Data restricted to married women, aged 25 or over, who were assigned to answer SRQ questions. Data were collected in the states of Bihar, Jharkhand, and Maharashtra.

Data source: SARI.

women who eat last have less education, on average, than women who do not eat last, the reweighting function puts more weight on more educated women who eat last and less weight on the less educated who eat last.

While the reweighting technique matches on the full distributions of educational attainment and asset wealth, allowing for flexible, non-parametric interaction between the two socioeconomic status variables, reweighting over many variables will partition the sample into many bins. This becomes problematic in instances when women who do not eat last have no counterparts in a given bin with women who do eat last, making the denominator in the reweighting function zero. For this analysis, when we reweight over the 24 education by asset ownership bins, there are no observations for which there were no matches, thus we do not need to drop any women who do not eat last from the sample.

**3.2.3. Ordered logistic regression.** In our second strategy, we use the parametric ordered logit regression to ask whether differences in mental health between women with lower and higher status are statistically significant, even accounting for socioeconomic status and other demographic characteristics. The ordered logistic regression approach allows us to control for a larger number of predictor variables than can be used in the reweighting described above.

Our outcome measure of mental health is the total number, out of 6 questions, that a respondent replied in the affirmative to experiencing symptoms associated with poor mental health. The outcome values of 0, 1, 2, 3, 4, 5, and 6 are the ordered categories that we model using a linear ordered logit regression model.

In this model, the values of the ordered categories of Total SRQ have a meaningful sequential order. However, underlying this variable is a latent variable $m^*$ that is an expanded version of the Total SRQ variable. For example, individuals may classify themselves into "yes" and "no" categories at varying levels of being able to think clearly in response to the question "Do you have trouble thinking clearly?" As respondents cross thresholds along a question's underlying spectrum, their values on the observed ordinal variable, Total SRQ, change. The cutpoints in the continuous distribution of $m^*$ that correspond to each ordered category in Total SRQ are fit by maximum likelihood. The ordered logistic regression allows us to estimate the effect of the independent variables on the underlying $m^*$ based on the assumption that the latent variable $m^*$ is a linear function of the independent variables. The error term in this model has a logistic distribution.

We write the linear model for $m^*$ as:

$$m_i^* = \beta_1 eatlast_i + \alpha_i\theta + E_i\lambda + A_i\varphi + \beta_5 Muslim_i + C_i\phi + S_i\omega + \varepsilon_i \qquad \text{(Eq 3)}$$

where $\varepsilon_i$ has a logistic distribution and the ordered logit link function additionally includes six cut-points for the seven levels of the outcome variable. Subscripts $i$ index respondents. The coefficient of interest is $\beta_1$ on eating last variable. We add in control variables in stages to see whether each predictor variables helps close the gap in mental health scores between women who eat last and those who do not. $\alpha_i$ is a set of dummy variables for the age of the respondent, in years; $E_i$ is a set of four indicators for educational attainment; $A_i$ is a set of six indicators for asset wealth; $Muslim_i$ is an indicator for being Muslim, $C_i$ is a set of six indicators for caste group; and $S_i$ is a set of three indicators for the respondent's state of residence. We additionally test a rural-only model that also includes an indicator variable, $latrine_i$, for whether the respondent has a latrine in her home or not.

The ordered logistic regression allows us to test a larger number of socioeconomic characteristics than the non-parametric reweighting technique to see if they can explain the differences in mental health between women who eat last and those who do not. However, one limitation of the ordered logit approach is that it assumes that the threshold or cut points of the latent variable are the same for all respondents, and that covariates have the same linear effect on the latent variable at every cut point.

### 3.3. Ethics statement

IRB approval for SARI data collection was obtained through the research institute for compassionate economics (r.i.c.e.) IRB, under protocol #16–003. Surveys were conducted by phone. Oral consent was obtained because surveyors did not meet the respondents in person. Consent was documented in Qualtrics software. Caste, religion, sex, age, and education categorizations were based on categorizations of the India Human Development Survey (2012). These categorizations were made for social scientific purposes and not required by any funder.

## 4. Results

Results are presented for ever-married women, aged 25 and above, in Bihar, Jharkhand, and Maharashtra, who were asked mental health questions using the Self-Reported Questionnaire. Estimates from all three states are pooled, and use weights to make the summary statistics in Table 3 representative of the populations of the three states as a whole.

### 4.1. Descriptive statistics

Table 3 shows the proportion of women who said that they had experienced the SRQ symptoms of having a lack of appetite (42%), having trouble sleeping (43%), having trouble thinking clearly (40%), feeling tired all of the time (19%), having trouble making decisions (65%), and having thought about committing suicide (44%) in the past 30 days. The left-most columns show proportions for ever-married women, aged 25 and over, in the sample, the middle columns show proportions for women who report eating last in their households, and the right most columns show proportions for women who report eating together with men in their households. Across all 6 symptoms, women who live in households where women eat last report higher rates of mental health distress than women who live in households where they eat with men.

Summary statistics for the independent variables used in the analysis are also shown in Table 3. The average age of respondents in the sample is approximately 40 years old, and respondents have an average of 4 years of education. 72% of respondents are from rural areas, 11% are Muslim, and approximately 25% of respondents are from underprivileged groups. The survey questions used the terms Dalit and Adivasi to refer to the constitutional groups Scheduled Castes and Scheduled Tribes, respectively. We use the terms "Dalit" and "Adivasi" in the paper to match the survey. We use household asset wealth as our indicator of economic status [46]: out of a total of 5 assets that SARI asked about, respondents have an average of 2.8 assets. We see that women who live in households where women eat last have less asset wealth and less education, on average. They are also more likely to have no education at all and are more likely to live in a rural place. Finally, rural women who report eating last are less likely to live in a home with a latrine than women who do not eat last.

In order to compare the two groups' mental health outcomes, a total SRQ score is calculated by adding up the number of symptoms, out of 6, for which a woman responded "yes" to experiencing the symptom in the past 30 days. Total SRQ scores range from zero to six, and a higher total SRQ score indicates worse mental health.

Fig 1 compares the weighted cumulative distribution functions of the total SRQ score distribution, between women who eat last and those who do not. The CDF of total SRQ for women who eat last is everywhere to the right of the CDF for women who do not eat last, indicating that women who eat last have a higher SRQ score, and thus worse mental health, than those who do not eat last.

Fig 2 below shows that at all asset levels (Panel A) and all levels of education (Panel B), the average total SRQ score for women who eat last is higher than for those who do not.

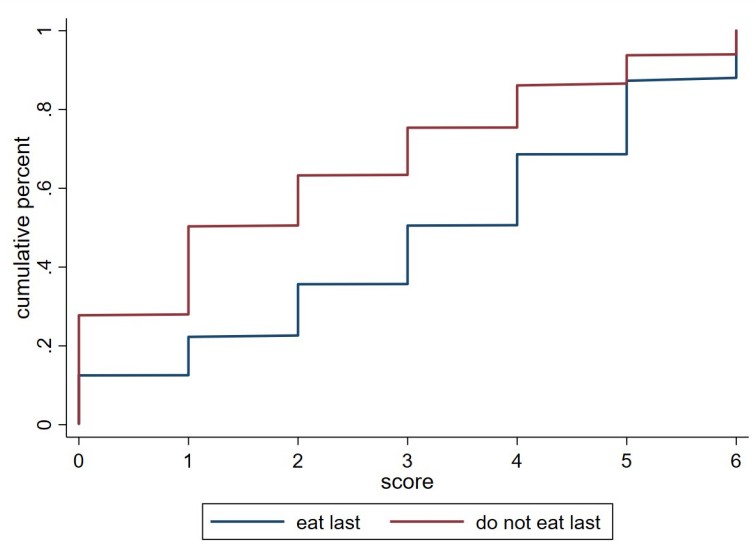

**Fig 1. CDF of SRQ score by eat last.** SARI data includes ever-married women, aged 25 or over, who were assigned to answer SRQ questions. Data were collected in the states of Bihar, Jharkhand, and Maharashtra. Women who eat last have a higher SRQ score, and thus worse mental health, than those who do not eat last. Data source: SARI.

## 4.2. Non-parametric reweighting and ordered logit regression

Next, we ask: to what extent gaps in mental health between women who eat last and those who do not can be explained by the differences in socioeconomic status between the two groups? We use two complementary strategies.

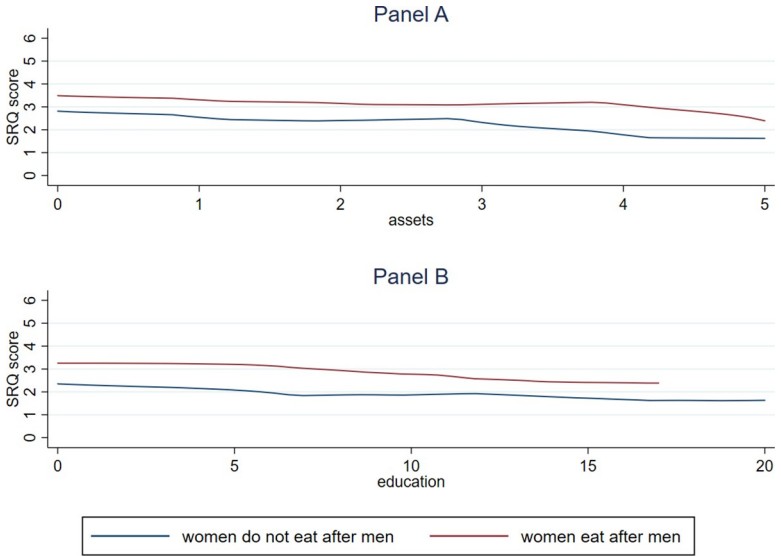

**Fig 2.** Mental health is worse among women who eat after men, at all levels of asset wealth (Panel A) and education (Panel B). SARI data includes ever-married women, aged 25 or over, who were assigned to answer SRQ questions. Data were collected in the states of Bihar, Jharkhand, and Maharashtra. Mental health, even accounting for wealth and education, is worse among women who face greater gender discrimination than women who do not. Data source: SARI.

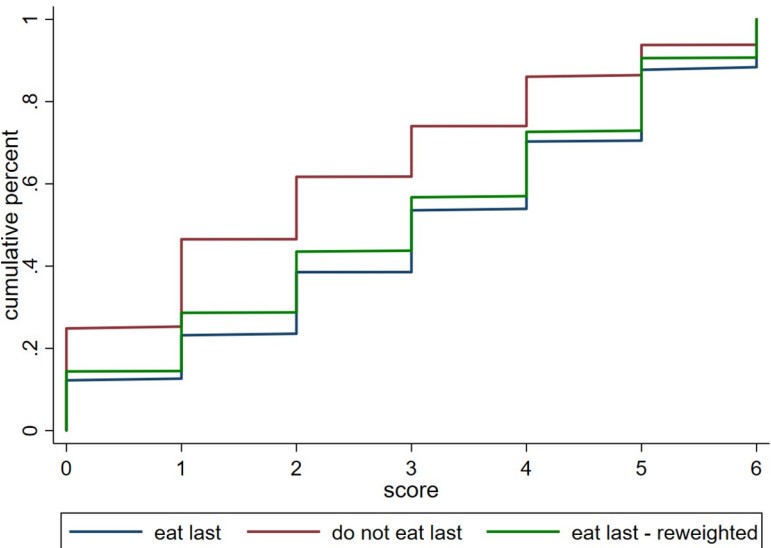

**Fig 3. Reweighted CDF by education and asset bins.** SARI data includes ever-married women, aged 25 or over, who were assigned to answer SRQ questions. Data were collected in the states of Bihar, Jharkhand, and Maharashtra. Although some of the difference in the distribution of mental health between women who do and do not eat last can be explained by differences in asset ownership and education, gaps in these observable characteristics cannot completely explain the difference in mental health outcomes between the two groups. Data source: SARI.

**4.2.1. Reweighting.** In Fig 3 below, we see that much of the difference in the distribution of mental health between women who do and do not eat last can be partly explained by differences in asset ownership and education. However, even after reweighting by these observable characteristics, a gap in mental health scores persists. Thus Fig 3 tells us that education and wealth differences between the two groups of women cannot completely explain the difference in mental health outcomes between them.

**4.2.2. Ordered logit regression.** Table 4 shows proportional odds from the ordered logit regressions of total SRQ on eating last and control variables. Models 1 through 7 include all respondents, and Model 8 includes only rural respondents. Model 1 includes no control variables, Model 2 controls for age categories, Model 3 additionally controls for education categories, Model 4 adds a control for being Muslim, Model 5 controls for the number of assets, Model 6 controls for caste group, and Model 7 controls for state dummies. Model 8, which include only rural residents, additionally controls for latrine ownership.

Across all models, eating last is able to predict mental health outcomes in this population: women who eat last have a statistically significantly higher proportional odds than women who do not eat last of having a higher SRQ score, or worse mental health. Model 1, with no controls, finds that women who eat last have 2.6 times the proportional odds of reporting worse mental health, compared to women who do not eat last. After adding the full set of socioeconomic controls, religion and caste, and state indicator variables reduces magnitude of the coefficient on eating last to about 1.9.

In Model 8, we restrict to only rural residents to test whether latrine ownership can explain some of the gap in mental health between the two groups. After controlling for the other variables in the model, household latrine ownership does not statistically significantly predict women's mental health. The coefficient on the indicator for eating last in Model 8 tells us that rural women who eat last have 2.4 times the proportional odds of having worse mental health than women who do not eat last.

**Table 4. Women who eat last have a greater proportional odds of reporting worse mental health, compared to women who do not eat last.**

| | SRQ total score | | | | | | | |
| | urban & rural | | | | | | | rural only |
| | (1) | (2) | (3) | (4) | (5) | (6) | (7) | (8) |
|---|---|---|---|---|---|---|---|---|
| Women eat last | 2.561*** | 2.567*** | 2.289*** | 2.285*** | 2.128*** | 2.190*** | 1.880* | 2.345*** |
| | (0.464) | (0.472) | (0.454) | (0.452) | (0.441) | (0.494) | (0.461) | (0.586) |
| Age categories (reference category: 25–34) | | | | | | | | |
| 35–44 | | 1.373 | 1.283 | 1.282 | 1.333 | 1.306 | 1.330 | 1.334 |
| | | (0.349) | (0.329) | (0.328) | (0.335) | (0.330) | (0.331) | (0.339) |
| 45–65 | | 1.161 | 0.988 | 0.997 | 1.145 | 1.098 | 1.140 | 1.143 |
| | | (0.252) | (0.231) | (0.234) | (0.289) | (0.279) | (0.286) | (0.308) |
| Education categories (reference category: 0 years) | | | | | | | | |
| 1–8 years | | | 0.600+ | 0.615+ | 0.770 | 0.774 | 0.799 | 0.971 |
| | | | (0.165) | (0.169) | (0.245) | (0.242) | (0.243) | (0.308) |
| 9–12 years | | | 0.556** | 0.565** | 0.862 | 0.865 | 0.898 | 0.957 |
| | | | (0.116) | (0.119) | (0.238) | (0.234) | (0.244) | (0.249) |
| more than 12 years | | | 0.529** | 0.535* | 0.995 | 0.975 | 0.987 | 0.622 |
| | | | (0.129) | (0.134) | (0.329) | (0.320) | (0.329) | (0.200) |
| Muslim | | | | 1.318 | 1.337 | 1.354 | 1.212 | 1.441 |
| | | | | (0.310) | (0.325) | (0.361) | (0.315) | (0.433) |
| Number of assets (reference category: 0 assets) | | | | | | | | |
| 1 | | | | | 0.724 | 0.782 | 0.802 | 0.873 |
| | | | | | (0.237) | (0.265) | (0.259) | (0.291) |
| 2 | | | | | 0.637 | 0.681 | 0.755 | 0.703 |
| | | | | | (0.239) | (0.259) | (0.273) | (0.285) |
| 3 | | | | | 0.744 | 0.839 | 0.973 | 1.290 |
| | | | | | (0.282) | (0.320) | (0.375) | (0.586) |
| 4 | | | | | 0.475 | 0.541 | 0.668 | 0.849 |
| | | | | | (0.228) | (0.251) | (0.309) | (0.424) |
| 5 | | | | | 0.321** | 0.360* | 0.444+ | 0.728 |
| | | | | | (0.137) | (0.151) | (0.191) | (0.353) |
| Caste group (reference group: Dalit) | | | | | | | | |
| OBC | | | | | | 0.773 | 0.716 | 0.599+ |
| | | | | | | (0.206) | (0.194) | (0.179) |
| General | | | | | | 0.802 | 0.789 | 1.168 |
| | | | | | | (0.201) | (0.195) | (0.320) |
| Brahmin | | | | | | 0.811 | 0.650 | 0.537 |
| | | | | | | (0.252) | (0.207) | (0.217) |
| Adivasi | | | | | | 0.498 | 0.520 | 0.387+ |
| | | | | | | (0.244) | (0.261) | (0.195) |
| Other | | | | | | 1.016 | 1.113 | 1.113 |
| | | | | | | (1.239) | (1.353) | (1.814) |
| State (reference group: Bihar) | | | | | | | | |
| Jharkhand | | | | | | | 1.035 | 1.295 |
| | | | | | | | (0.215) | (0.330) |
| Maharashtra | | | | | | | 0.608+ | 0.654 |
| | | | | | | | (0.155) | (0.182) |
| Latrine | | | | | | | | 0.695 |
| | | | | | | | | (0.172) |

(*Continued*)

**Table 4.** (Continued)

| | SRQ total score | | | | | | | |
| | urban & rural | | | | | | | rural only |
| | (1) | (2) | (3) | (4) | (5) | (6) | (7) | (8) |
| n | 1144 | 1144 | 1144 | 1143 | 1139 | 1111 | 1111 | 777 |

Note: Standard errors in parentheses.

+ p<0.1

* p<0.05

** p<0.01

*** p<0.001. Models 1–7 restricted to married women, aged 25 or over, who were assigned to answer SRQ questions. Data were collected in the states of Bihar, Jharkhand, and Maharashtra.

Model 8 additionally restricted to rural residents only. Data source: SARI.

In S2 Table, we break our sample down into caste groups to better understand whether the association between gender discrimination and mental health that we see in Table 4 differs by caste group (for the full urban and rural sample). We see that among OBC women, those who eat last have 2.1 times the proportional odds of reporting worse mental health, even controlling for socioeconomic stats, religion, caste, and state. Unfortunately, our sample sizes become too small to be able to find statistically significant results for the other caste groups, but the coefficients have the expected sign.

## 4.3. Does autonomy mediate the relationship between eating last and mental health?

In this section, we explore one possible pathway through which eating last may be associated with poor mental health among women: through the psychological stress that might result from a lack of autonomy. We examine two separate measures. The first is from SARI, which measured autonomy by asking respondents if they were allowed to go to a neighbors' house without asking for permission. The second is from the IHDS, which measured autonomy using questions related to women's decision-making power.

**4.3.1. Autonomy in SARI: Ability to leave the house without permission.** In order to test whether eating last is associated with poor mental health because the lack of autonomy and power within the household, we add a measure of lack of autonomy to the ordered logit regression described in Section 4.2.2, and assess whether including this measure of autonomy reduces or eliminates the coefficient on eating last. The measure of autonomy that we use is whether a woman can leave the house without permission. In particular, SARI asked women: "When you want to go outside alone somewhere near your home, such as to visit a neighbor, do you need to ask your husband or family, or do you just tell them and go?"

Column 1 of Table 5 is reproduced from Column 1 of Table 4, to show the relationship between eating last and mental health. Column 2 of Table 5 shows results of an ordered logit regression of SRQ score on eating last, controlling for having to ask for permission to go out. We see that controlling for having to ask for permission to go out does not change the statistical significance of eating last on total SRQ.

Column 3 of Table 5 is reproduced from Column 7 of Table 4, showing the relationship between eating last and mental health with controls for age, education, being Muslim, assets, caste, and state dummies. Column 4 of Table 5 shows ordered logit regression results of SRQ score on eating last, additionally controlling for having to ask for permission to go out. We find that even with the full set of controls and controlling for asking for permission, the

**Table 5. Women's autonomy does not clearly mediate the relationship between mental health and eating last.**

| | SRQ total score | | | |
| | urban & rural | | | |
| | **(1)** | **(2)** | **(3)** | **(4)** |
|---|---|---|---|---|
| Women eat last | 2.561*** | 2.578*** | 1.880* | 1.891** |
| | (0.464) | (0.472) | (0.461) | (0.464) |
| Ask for permission to go to neighbor's house | | 0.944 | | 0.955 |
| | | (0.182) | | (0.190) |
| Age categories (reference category: 25–34) | | | | |
| 35–44 | | | 1.330 | 1.325 |
| | | | (0.331) | (0.325) |
| 45–65 | | | 1.140 | 1.132 |
| | | | (0.286) | (0.292) |
| Education categories (reference category: 0 years) | | | | |
| 1–8 years | | | 0.799 | 0.803 |
| | | | (0.243) | (0.242) |
| 9–12 years | | | 0.898 | 0.898 |
| | | | (0.244) | (0.243) |
| more than 12 years | | | 0.987 | 0.981 |
| | | | (0.329) | (0.330) |
| Muslim | | | 1.212 | 1.218 |
| | | | (0.315) | (0.318) |
| Number of assets (reference category: 0 assets) | | | | |
| 1 | | | 0.802 | 0.798 |
| | | | (0.259) | (0.256) |
| 2 | | | 0.755 | 0.746 |
| | | | (0.273) | (0.262) |
| 3 | | | 0.973 | 0.973 |
| | | | (0.375) | (0.374) |
| 4 | | | 0.668 | 0.662 |
| | | | (0.309) | (0.306) |
| 5 | | | 0.444+ | 0.439+ |
| | | | (0.191) | (0.187) |
| Caste group (reference group: Dalit) | | | | |
| OBC | | | 0.716 | 0.714 |
| | | | (0.194) | (0.192) |
| General | | | 0.789 | 0.790 |
| | | | (0.195) | (0.196) |
| Brahmin | | | 0.650 | 0.648 |
| | | | (0.207) | (0.205) |
| Adivasi | | | 0.520 | 0.521 |
| | | | (0.261) | (0.261) |
| Other | | | 1.113 | 1.126 |
| | | | (1.353) | (1.360) |
| State (reference group: Bihar) | | | | |
| Jharkhand | | | 1.035 | 1.034 |
| | | | (0.215) | (0.215) |
| Maharashtra | | | 0.608+ | 0.612+ |
| | | | (0.155) | (0.159) |

(*Continued*)

**Table 5.** (Continued)

| | SRQ total score | | | |
| | urban & rural | | | |
| | (1) | (2) | (3) | (4) |
| Latrine | | | | |
| n | 1144 | 1144 | 1111 | 1111 |

Note: Standard errors in parentheses.

+ p<0.1

* p<0.05

** p<0.01

*** p<0.001. all models restricted to married women, aged 25 or over, who were assigned to answer SRQ questions. Data were collected in the states of Bihar, Jharkhand, and Maharashtra.

statistically significant relationship between eating last and mental health remains robust. While this suggests that autonomy may not be an important pathway through which eating last is associated with mental health, this is not definitive evidence, because it is possible that other indicators of autonomy would mediate the relationship. We discuss this in more detail in the Discussion, and consider another indicator of autonomy in the following section.

**4.3.2. Autonomy in the IHDS: Decision-making power.**   Next, we report on analyses of the IHDS data that are suggestive of what might happen if we could run a regression of mental health on eating last and possible mediators, even though such a regression is not actually possible because the IHDS did not collect mental health (and SARI did not collect decision-making power). In particular, we explore whether women's decision-making power is correlated with the likelihood of women eating last in their households.

The IHDS asks women a series of questions about personal and household decision making for a variety of decisions. The first question asks "Please tell me who in your family has the most say in what to cook on a daily basis," with options for the respondent herself, and other members of the household. Subsequent questions ask about decision-making with regard to buying expensive items, the number of children to have, what to do if the respondent herself gets sick, buying land or property, how much money to spend for a social function, what to do if a child gets sick (for women who have children), and how to arrange children's marriages.

If the respondent reported that she had the most say in a particular type of decision, the answer was coded as 1, and answers that indicated someone else in the household had the most say in that decision were coded as 0. We then added up the number of decisions in which the respondent had the most say, for a total between 0 and 8. A total of 0 indicates either no or a very low level of decision-making power, and a total of 8 indicates a high level of decision-making power. This total was broken up into categories of 0 total decisions in which a woman reports having the most say, 1–2 decisions, 3–4 decisions, 5–6 decisions, and 7–8 decisions.

To test whether decision-making power predicts eating last, we run OLS regressions of the following form:

$$eatlast_i = \beta_1\, 1-2\ decisions_i + \beta_2\, 3-4\ decisions_i + \beta_3\, 5-6\ decisions_i + \beta_4\, 7-8\ decisions_i + \alpha_i\theta + E_i\lambda + A_i\varphi + \beta_5 Muslim_i + C_i\phi + S_i\omega + \varepsilon_i \quad (\text{Eq 4})$$

Where $i$ indexes individual women. The coefficients of interest are $\beta_1$, $\beta_2$, $\beta_3$, and $\beta_4$ which represent the categories of the number of decisions a woman has the most say in, compared to a reference category of having the most say in 0 decisions. We also include controls to see whether decision-making predicts mental health, net of age, education, wealth, religion, caste,

and state: $\alpha_i$ is a set of dummy variables for the age of the respondent, in years; $E_i$ is a set of four indicators for educational attainment; $A_i$ is a set of six indicators for asset wealth; $Muslim_i$ is an indicator for being Muslim, $C_i$ is a set of six indicators for caste group; and $S_i$ is a set of three indicators for the respondent's state of residence.

Table 6 shows that women who have greater decision making power in their households are less likely to eat last. After including controls for age category, education category, asset quintile, being Muslim, caste category, and state dummies, women who have the most say over

**Table 6. Women who report greater decision-making power are less likely to eat last in their households (IHDS).**

| | Women eat last | |
|---|---|---|
| | (1) | (2) |
| Decision-making power (reference category: 0 decisions) | | |
| 1–2 decisions | -0.205*** | -0.0697*** |
| | (0.0179) | (0.0171) |
| 3–4 decisions | -0.217*** | -0.140*** |
| | (0.0240) | (0.0220) |
| 5–6 decisions | -0.316*** | -0.161*** |
| | (0.0387) | (0.0359) |
| 7–8 decisions | -0.389*** | -0.257*** |
| | (0.0336) | (0.0313) |
| Age categories (reference category: 25–34) | | |
| 35–44 | | 0.00509 |
| | | (0.0158) |
| 45–65 | | 0.0219 |
| | | (0.0173) |
| Education categories (reference category: 0 years) | | |
| 1–8 years | | 0.00473 |
| | | (0.0177) |
| 9–12 years | | -0.00460 |
| | | (0.0213) |
| more than 12 years | | -0.100* |
| | | (0.0417) |
| Muslim | | -0.0457+ |
| | | (0.0235) |
| Asset quintiles (reference category: Poorest) | | |
| 2nd quintile | | 0.0536** |
| | | (0.0202) |
| Middle | | 0.0144 |
| | | (0.0219) |
| 4th quintile | | -0.0500* |
| | | (0.0248) |
| Richest | | -0.00817 |
| | | (0.0277) |
| Caste group (reference group: Dalit) | | |
| OBC | | -0.0243 |
| | | (0.0181) |
| General | | 0.0592** |
| | | (0.0207) |

(*Continued*)

**Table 6.** (Continued)

| | Women eat last | |
| --- | --- | --- |
| | (1) | (2) |
| Brahmin | | 0.0563 |
| | | (0.0421) |
| Adivasi | | -0.0315 |
| | | (0.0269) |
| State (reference group: Bihar) | | |
| Jharkhand | | -0.00975 |
| | | (0.0212) |
| Maharashtra | | -0.386*** |
| | | (0.0188) |
| n | 3772 | 3772 |

Note: Standard errors in parentheses.

+ p<0.1

* p<0.05

** p<0.01

*** p<0.001. Both models include ever-married women, aged 25 or over, in the states of Bihar, Jharkhand, and Maharashtra. Data source: IHDS.

1–2 decisions are 7.0 percentage points less likely to eat last than women have the most say over 0 decisions; women who have the most say in 3–4 decisions are 14.0 percentage points less likely to eat last; women who have the most say in 5–6 decisions are 16.1 percentage points less likely to eat last; and women who have the most say in 7–8 decisions are 25.7 percentage points less likely to eat last.

## 5. Discussion

Although prior studies find associations between women's social status and mental health across a variety of contexts, this study is the first to explore this relationship in population data from India. We find consistent results that one measure of discrimination against women–living in a household in which women eat only after the men have eaten–is correlated with worse self-reported mental health as measured by an adapted Self-Reporting Questionnaire.

In India, Coffey et al. [8] find evidence that eating last is associated with worse physical health. Using IHDS data to analyze the difference in the fraction of women who are underweight between households in which men eat first and households in which men do not eat first, they find that women who experience this particular type of gender discrimination are more likely to be underweight than women who do not face this type of discrimination, at all levels of household expenditure. The robust association between eating last and underweight suggests that eating last may have an impact on physical health, which in turn, may have an impact on mental health. Similarly, evidence from the US finds that food insufficiency is strongly associated with women's self-reported depression [47]. One mechanism they explore is that food insufficiency leads to lower nutrient intakes, which could negatively impact immunity and thus increase the risk of a variety of chronic diseases. Other studies have shown that across countries, mental health disorders and chronic physical conditions commonly occur together [19].

Our analysis of two measures of autonomy–being able to go out without permission and decision-making power–suggest that while autonomy may mediate the relationship between gender discrimination and mental health, some measures of autonomy may be better able to capture this. It may seem puzzling that asking for permission does not seem to be a statistically significant mediator, while the association between decision-making and eating last is quite strong. S3 Table uses SARI data to show the relationship between asking for permission and eating last, and finds that asking for permission is associated with a *lower* likelihood of eating last. This suggests that asking for permission may not, in fact, be an adequate measure for capturing women's autonomy. It is possible that asking for permission may simply be a courtesy that women perform, even knowing that, in fact, they have the freedom to go out if they choose. Households in which more courtesy is expected may also be those in which women are less likely to eat last. This is a possible reason why asking permission to leave the house does not mediate the relationship between eating last and mental health outcomes. Decision-making may simply be a more accurate measure of women's status.

This study has shown how one form of gender discrimination–in particular, women having to wait to eat until the men in the household have finished eating–may have deleterious effects on women's mental health. This is an important finding because poor mental health among women impacts them as individual and can also impact the well-being of their children [48, 49]. It is possible that the relationship between eating last and poor mental health is working both through women's physical health and a lack of autonomy. However, we are cautious in interpreting the results on lack of autonomy until further data on autonomy, mental health, and the practice of eating last are collected. Future research should collect data on a wide range of measures of women's autonomy and social status, in conjunction with measures of physical and mental health. This will move us towards a more nuanced understanding of how gender discrimination and autonomy in patriarchal societies impact women's mental wellbeing.

## Supporting information

**S1 Table. Complete SRQ questionnaire (those used in SARI are marked with an asterisk).**
(PDF)

**S2 Table. Association between eating last and mental health in SARI, by caste group.**
(PDF)

**S3 Table. Women who report having to ask for permission to go a neighbor's house are less likely to eat last in their households.**
(PDF)

## Acknowledgments

We wholeheartedly thank Nidhi Khurana for expertly guiding the SARI survey team. We also thank the SARI interviewers, without whose hard work and dedication these ideas could not have been implemented: Bharati Kadam, Dilip Singh Shekawat, Gunjan Kumari, Kailash Kumar, Kavita Naik, Krishna Maruti Narer, Laxmi Saini, Nasima Shaikh, Nisarg Jagtap, Poonam Saini, Pragati Pandurang Desai, Pramod Rajak, Pranav Narayan Lawande, Priyanka Singh, Ramita Karna, Rohini Hanbartii, Sachin Shere, Sanita Sutar, Sharmili Karkar, Tawsin Gulab Mulla, and Vivek Koli.

## Author Contributions

**Conceptualization:** Payal Hathi, Diane Coffey, Amit Thorat, Nazar Khalid.

**Data curation:** Payal Hathi, Diane Coffey, Nazar Khalid.

**Formal analysis:** Payal Hathi, Diane Coffey.

**Investigation:** Payal Hathi, Diane Coffey, Nazar Khalid.

**Methodology:** Payal Hathi, Diane Coffey.

**Project administration:** Payal Hathi, Diane Coffey, Amit Thorat, Nazar Khalid.

**Writing – original draft:** Payal Hathi.

**Writing – review & editing:** Payal Hathi, Diane Coffey, Amit Thorat, Nazar Khalid.

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
