## [Decision Letter · Decision Letter 0]

20 Nov 2020

PONE-D-20-26210

When women eat last: Discrimination at home and women's mental health

PLOS ONE

Dear Dr. Hathi,

Thank you for submitting your manuscript to PLOS ONE. After careful consideration, we feel that it has merit but does not fully meet PLOS ONE’s publication criteria as it currently stands. Therefore, we invite you to submit a revised version of the manuscript that addresses the points raised during the review process.

We look forward to receiving your revised manuscript.

Kind regards,

Kannan Navaneetham, PhD

Academic Editor

PLOS ONE

Journal Requirements:

2. We note you have included a table to which you do not refer in the text of your manuscript. Please ensure that you refer to Table 3 in your text; if accepted, production will need this reference to link the reader to the Table.

Reviewers' comments:

Reviewer's Responses to Questions

**Comments to the Author**

1. Is the manuscript technically sound, and do the data support the conclusions?

Reviewer #1: Yes

Reviewer #2: Yes

2. Has the statistical analysis been performed appropriately and rigorously? 

Reviewer #1: Yes

Reviewer #2: Yes

3. Have the authors made all data underlying the findings in their manuscript fully available?

Reviewer #1: Yes

Reviewer #2: Yes

4. Is the manuscript presented in an intelligible fashion and written in standard English?

Reviewer #1: Yes

Reviewer #2: Yes

5. Review Comments to the Author

Reviewer #1: Comments on Manuscript PONE-D-20-26210

This is an important and interesting paper that investigates whether women eating last in the household (a widespread practice in India) affects their mental health. The authors use a unique dataset to show that it indeed affects women’s mental health, and the relationship is robust to controlling for various observable characteristics such as education and asset. The authors also discuss other potential pathways such as women’s physical health and autonomy, although they cannot provide any direct evidence that would adequately establish these pathways. I have the following comments that the authors may consider in revising their paper.

1. It will be good to explore heterogeneity, especially on the intersectionality aspect considering caste. Various studies in the literature highlight the intersectionality (or the lack thereof) between gender and caste in the Indian context. So, the authors can estimate the full model separately for different castes to see whether the effect varies by caste (alternatively interaction of “women eat last” with caste dummies can be used).

2. I find it a bit strange that the autonomy variable included in Table 5 neither has any mediating effect nor it is significant itself. The authors do suggest that this measure may not be adequately capturing autonomy. Therefore, the authors investigate the effect of an alternative measure of autonomy from IHDS data and show that autonomy in that data strongly predicts the probability of eating last for the women. However, since IHDS do not have measure on mental health, they cannot investigate that relationship. Rather, they use the evidence from IHDS as a basis for discussion of their original result using SARI data.

In my view, this analysis and the discussion are incomplete. In the regression presented in Table 5 (using SARI data), the measure of autonomy (i.e., “Ask for permission to go to neighbor’s house” variable) will be a significant mediator only if it is correlated with both “Women eat last” variable and the outcome variable (i.e. mental health). The autonomy variable itself is not a significant predictor of the outcome variable – this is visible from the results presented in the table. But is the autonomy variable also uncorrelated with “Women eat last” variable? That evidence is not presented. In other words, the authors could run a regression using the SARI data itself, where “Women eat last” is the dependent variable and autonomy is an explanatory variable. This is analogous to the IHDS regression presented in Table 6. Then results from this regression using SARI data could be put in contrast with the regression using IHDS data. If it happens to be the case that SARI measure of autonomy is not significant, while the IHDS measure of autonomy is significant (as seen from Table 6), then it would indicate more clearly that the SARI measure is inadequate to capture women’s autonomy. On the other hand, if SARI measure also comes out significant just like IHDS measure, then the interpretation will be different.

3. On line 621, it should be Table 6, not Table 5.

4. The paper does not mention in which year the SARI data was collected.

Reviewer #2: This an important piece of work highlighting effect of gender discrimination, manifested as practice of women eating last, on mental health and well-being. I have few minor concerns and require some revision -

1. in line 261, 6 questions are selected from 20 without providing process and validity. Suggest adding both these in the method section.

2. In line 321, you have considered latrine ownership only in case of rural areas. While I am fine with the assumption that open defecation is practiced mainly in rural areas but access to safe sanitation facilities are major concern in urban areas with high population density and limited open space, particularly for those living in slum areas

3. Check table numbers mentioned in results section

4. In Table 3 - summary statistics - mean is referred for proportion of women who self-reported mental health symptoms in past 30 days. Need correction

5. Any specific reason for keeping Brahmin separate from general caste group? Also, Dalit and Adivasi are not constitutional terms. Suggest keeping it Scheduled Castes and Scheduled Tribes and mention their underprivileged condition for wider understanding. Also, avoid using upper caste and lower caste. These are privileged and underprivileged caste/tribe groups.

6. Measure on mobility is inadequate. Any standard population survey captures mobility to 4-5 location within vicinity and outside the community. Any reason for considering this in the study? You haven’t included this in Table 3.

7. I did not see any value of adding analysis of data on decision making from IHDS.

6. PLOS authors have the option to publish the peer review history of their article (what does this mean?). If published, this will include your full peer review and any attached files.

Reviewer #1: **Yes: **Soham Sahoo

Reviewer #2: **Yes: **Pranita Achyut

---

## [Author Response · Author response to Decision Letter 0]

1 Jan 2021

Dear PLoS One Editors, 

Thank you for the opportunity to revise our paper, “When women eat last: Discrimination at home and women's mental health” (PONE-D-20-26210) for resubmission to PLoS One. The reviewers made several helpful suggestions to improve the paper. Below (and in the attached 'Response to Reviewers', we describe how we have incorporated their feedback in response to each of their comments. For ease of re-review, reviewers’ comments are reproduced below in bold and indented, and our responses are in lightweight, indented text. 

Reviewer #1: Comments on Manuscript PONE-D-20-26210

This is an important and interesting paper that investigates whether women eating last in the household (a widespread practice in India) affects their mental health. The authors use a unique dataset to show that it indeed affects women’s mental health, and the relationship is robust to controlling for various observable characteristics such as education and asset. The authors also discuss other potential pathways such as women’s physical health and autonomy, although they cannot provide any direct evidence that would adequately establish these pathways. I have the following comments that the authors may consider in revising their paper.

1. It will be good to explore heterogeneity, especially on the intersectionality aspect considering caste. Various studies in the literature highlight the intersectionality (or the lack thereof) between gender and caste in the Indian context. So, the authors can estimate the full model separately for different castes to see whether the effect varies by caste (alternatively interaction of “women eat last” with caste dummies can be used).

We agree that this would be an interesting addition to the paper. In a new supplementary appendix table (Table S2), we have conducted the main analysis (the association between eating last and mental health) for each caste group separately, including the full set of covariates. For each caste group, the direction of the coefficient is what we would expect (women who eat last report worse mental health), but results are only statistically significant for OBC women, which is the largest caste group in the sample. We suspect that sample sizes for the other groups are too small to be able to detect statistically significant results. We have added text to explain this finding. 

2. I find it a bit strange that the autonomy variable included in Table 5 neither has any mediating effect nor it is significant itself. The authors do suggest that this measure may not be adequately capturing autonomy. Therefore, the authors investigate the effect of an alternative measure of autonomy from IHDS data and show that autonomy in that data strongly predicts the probability of eating last for the women. However, since IHDS do not have measure on mental health, they cannot investigate that relationship. Rather, they use the evidence from IHDS as a basis for discussion of their original result using SARI data.

In my view, this analysis and the discussion are incomplete. In the regression presented in Table 5 (using SARI data), the measure of autonomy (i.e., “Ask for permission to go to neighbor’s house” variable) will be a significant mediator only if it is correlated with both “Women eat last” variable and the outcome variable (i.e. mental health). The autonomy variable itself is not a significant predictor of the outcome variable – this is visible from the results presented in the table. But is the autonomy variable also uncorrelated with “Women eat last” variable? That evidence is not presented. In other words, the authors could run a regression using the SARI data itself, where “Women eat last” is the dependent variable and autonomy is an explanatory variable. This is analogous to the IHDS regression presented in Table 6. Then results from this regression using SARI data could be put in contrast with the regression using IHDS data. If it happens to be the case that SARI measure of autonomy is not significant, while the IHDS measure of autonomy is significant (as seen from Table 6), then it would indicate more clearly that the SARI measure is inadequate to capture women’s autonomy. On the other hand, if SARI measure also comes out significant just like IHDS measure, then the interpretation will be different.

We have added a new supplementary appendix table (Table S3) showing regression results of the association between asking for permission to leave the house and eating last, using SARI data. We find that having to ask for permission to leave the house decreases the proportional odds of eating last. This goes in the opposite direction of what we would expect if asking for permission is a sign of autonomy, helping us demonstrate that may not be an adequate measure to capture women's autonomy. During debriefing with interviewers during and after the survey, some said that they believed that this questions may have been capturing a courtesy that women paid to other members of the household, rather than a marker of their freedom of movement. This interpretation is consistent with the association in Table S3. Households in which people pay one another more courtesy would also be those in which women may be less likely to eat last. We have added text to explain this finding.

3. On line 621, it should be Table 6, not Table 5.

Thank you! This error has been corrected. 

4. The paper does not mention in which year the SARI data was collected.

Years have been added into Table 1, and in the text.

Reviewer #2: 

This an important piece of work highlighting effect of gender discrimination, manifested as practice of women eating last, on mental health and well-being. I have few minor concerns and require some revision –

1. in line 261, 6 questions are selected from 20 without providing process and validity. Suggest adding both these in the method section.

The SARI questionnaire included the SRQ mental health questions as part of an experiment testing the feasibility of asking questions about mental health by phone. The SRQ questions were tested against a questionnaire called the Kessler-6, which includes six questionf focused on emotional symptoms of mental health distress. The six SRQ questions were selected to focus on physical symptoms, as a contrast to the Kessler-6 questions. They were tested during questionnaire piloting in late 2017 and early 2018, to ensure that respondents understood what was being asked. We have added more information about this process and validation to the text, and also cited a paper which describes the mental health questions in greater detail.

2. In line 321, you have considered latrine ownership only in case of rural areas. While I am fine with the assumption that open defecation is practiced mainly in rural areas but access to safe sanitation facilities are major concern in urban areas with high population density and limited open space, particularly for those living in slum areas

We agree that sanitation is also an important issue in urban areas, however, India's open defecation is largely concentrated in rural areas. In addition, the Swachh Bharat Mission was largely focused on rural areas, so we wanted to get a sense of how that policy was working, particularly in terms of latrine construction and use. Another reason for only asking about latrine ownership and use in rural is now provided in the text: "...we did not ask urban residents about latrine ownership and use was that, in piloting, they sometimes became upset at the suggestion that they might defecate in the open."

3. Check table numbers mentioned in results section

We have fixed table numbers for Table 3 and Table 6, and references to those tables in the text.

4. In Table 3 – summary statistics – mean is referred for proportion of women who self-reported mental health symptoms in past 30 days. Need correction

We have added in "proportion" to the title of the column to clarify.

5. Any specific reason for keeping Brahmin separate from general caste group? Also, Dalit and Adivasi are not constitutional terms. Suggest keeping it Scheduled Castes and Scheduled Tribes and mention their underprivileged condition for wider understanding. Also, avoid using upper caste and lower caste. These are privileged and underprivileged caste/tribe groups.

Thank you for this. We have disaggregated caste groups as much as possible, and as Brahmins have particular privilege in the states we study, we wanted to ask about them separately. Although the constitutional terms are Scheduled Caste and Scheduled Tribe, we have used the terms "Dalit" and "Adivasi" in the paper as these are the terms that were used in the survey. We chose them for the survey because they were more familiar to the respondents, many of whom were women with low levels of education. We have added a clarification to the text that Dalit and Adivasi correspond to SC and ST respectively. Thank you for the suggestion about removing "lower caste." We agree -- we have removed this reference.

6. Measure on mobility is inadequate. Any standard population survey captures mobility to 4-5 location within vicinity and outside the community. Any reason for considering this in the study? You haven’t included this in Table 3.

Unfortunately, due to the time constraints of a phone survey, we were unable to include additional question capturing measures of women's mobility in SARI (just the one about whether she can go to her neighbor's house on her own or not). If we had been able to collect this data, we would have included it, but instead, we included questions on other manifestations of gender inequality. 

7. I did not see any value of adding analysis of data on decision making from IHDS.

The purpose of the analysis on decision making from the IHDS was to explore whether any other measures of autonomy, that we were not able to collect data on in SARI, could help us explain the connection between poor mental health and eating last. We have included a supplementary analysis of the association between SARI's measure of autonomy (asking for permission to go to a neighbor's house) and eating last, as a parallel to the IHDS analysis, which should make clear the purpose of looking at two different measures of autonomy.

---

## [Decision Letter · Decision Letter 1]

1 Feb 2021

When women eat last: Discrimination at home and women's mental health

PONE-D-20-26210R1

Dear Dr. Hathi,

We’re pleased to inform you that your manuscript has been judged scientifically suitable for publication and will be formally accepted for publication once it meets all outstanding technical requirements.

Kind regards,

Kannan Navaneetham, PhD

Academic Editor

PLOS ONE

Additional Editor Comments (optional):

Reviewers' comments:

Reviewer's Responses to Questions

**Comments to the Author**

1. If the authors have adequately addressed your comments raised in a previous round of review and you feel that this manuscript is now acceptable for publication, you may indicate that here to bypass the “Comments to the Author” section, enter your conflict of interest statement in the “Confidential to Editor” section, and submit your "Accept" recommendation.

Reviewer #1: All comments have been addressed

Reviewer #2: All comments have been addressed

2. Is the manuscript technically sound, and do the data support the conclusions?

Reviewer #1: Yes

Reviewer #2: Yes

3. Has the statistical analysis been performed appropriately and rigorously? 

Reviewer #1: Yes

Reviewer #2: Yes

4. Have the authors made all data underlying the findings in their manuscript fully available?

Reviewer #1: (No Response)

Reviewer #2: Yes

5. Is the manuscript presented in an intelligible fashion and written in standard English?

Reviewer #1: Yes

Reviewer #2: Yes

6. Review Comments to the Author

Reviewer #1: (No Response)

Reviewer #2: (No Response)

7. PLOS authors have the option to publish the peer review history of their article (what does this mean?). If published, this will include your full peer review and any attached files.

Reviewer #1: **Yes: **Soham Sahoo

Reviewer #2: No

---

## [Editor Report · Acceptance letter]

19 Feb 2021

PONE-D-20-26210R1 

When women eat last: Discrimination at home and women’s mental health 

Dear Dr. Hathi:

I'm pleased to inform you that your manuscript has been deemed suitable for publication in PLOS ONE. Congratulations! Your manuscript is now with our production department. 

Kind regards, 

on behalf of

Professor Kannan Navaneetham 

Academic Editor

PLOS ONE